# Antiviral Agents for Preventing Cytomegalovirus Disease in Recipients of Hematopoietic Cell Transplantation

**DOI:** 10.3390/v16081268

**Published:** 2024-08-08

**Authors:** Tang-Her Jaing, Yi-Lun Wang, Chia-Chi Chiu

**Affiliations:** 1Division of Hematology and Oncology, Department of Pediatrics, Chang Gung Memorial Hospital, Taoyuan 33315, Taiwan; g987669@gmail.com; 2Division of Nursing, Chang Gung Memorial Hospital, Taoyuan 33315, Taiwan; chi0105@cgmh.org.tw

**Keywords:** antiviral agents, allogeneic hematopoietic cell transplantation, cytomegalovirus, pediatric

## Abstract

This systematic review discusses the use of prophylaxis to prevent cytomegalovirus (CMV) infection in recipients who have undergone hematopoietic cell transplantation. It highlights the need for new approaches to control and prevent CMV infection. The approval of the anti-CMV drug letermovir has made antiviral prophylaxis more popular. CMV-specific T cell-mediated immunity tests are effective in identifying patients who have undergone immune reconstitution and predicting disease progression. Maribavir (MBV) has been approved for the treatment of post-transplant CMV infection/disease in adolescents. Adoptive T-cell therapy and the PepVax CMV vaccine show promise in tackling refractory and resistant CMV. However, the effectiveness of PepVax in reducing CMV viremia/disease was not demonstrated in a phase II trial. Cell-mediated immunity assays are valuable for personalized management plans, but more interventional studies are needed. MBV and adoptive T-cell therapy are promising treatments, and trials for CMV vaccines are ongoing.

## 1. Introduction

### 1.1. Cytomegalovirus Overview

Human cytomegalovirus (CMV) infects 60% of adults in developed countries and 90% in developing countries. Despite its prevalence, this infection typically manifests as asymptomatic or pauci-symptomatic. Nevertheless, it can result in a variety of complications, including sensorineural hearing loss, in immunocompromised individuals or infants with congenital infections [1]. The review focuses on clinical CMV in children with secondary immunodeficiency, specifically pediatric cancer patients and children undergoing hematopoietic cell transplantation (HCT). CMV immunotherapy is one of the highlighted treatments for CMV [2]. CMV disease remains a major cause of illness and death in individuals who have undergone HCT, with mortality rates reaching as high as 60%. CMV infection is linked to a higher likelihood of developing secondary infections, graft-versus-host disease (GVHD), and non-relapse mortality (NRM). Risk factors may include the use of high-dose corticosteroids, T-cell depletion, GVHD, and donors who are not a perfect match. Monitoring immune factors can aid in understanding the pathogenesis of CMV reactivation and making informed clinical decisions [3,4]. The objective of this study was to conduct a comprehensive review of the latest antiviral agents by analyzing the current literature.

### 1.2. Risk Factors for CMV Viremia after HCT

The outcomes after Allo-HCT are significantly influenced by the CMV serostatus, with the greatest impact observed in recipients who test positive. CMV seronegative donor/recipient pairs exhibit the highest survival rates [5]. There have been studies indicating that CMV seropositive patients may experience better outcomes when receiving a transplant from a seropositive donor compared to those who receive a transplant from a CMV seronegative donor [6]. Some risk factors to consider are T cell depletion, high-dose steroids, HLA mismatched donors, and GVHD. The use of myeloablative conditioning regimens can lead to T-cell dysfunction, which in turn increases the likelihood of CMV reactivation in patients with GVHD. The use of steroids can have a negative impact on the immune system, potentially leading to an increased risk of bacterial and CMV infections due to the delayed immune reconstitution in umbilical cord blood transplants [7].

A study examined 6968 patients with chronic hematological malignancies who received allogeneic HCT. It was discovered that the presence of CMV in both the donor and recipient was linked to a notable decrease in overall survival (OS) among acute leukemia patients [8]. The study concluded that CMV seropositivity had a significant impact on progression-free survival (PFS), OS, and NRM, regardless of other factors. CMV-seropositive recipients with a CMV-seronegative donor experienced the most significant decline in OS. According to the study, Allo-HCT patients may experience a decline in overall survival due to CMV seropositivity in both the donor and recipient. Consideration should be given to the reactivation of CMV due to several factors. These factors include the CMV seropositivity of both the recipient and the donor, the level of immune suppression in the recipient, and differences in human leukocyte antigen between the donor and recipient [9]. Myeloablative conditioning regimens are more cytotoxic compared to reduced-intensity or non-myeloablative regimens. However, both types of regimens lead to T-cell dysfunction [10]. CMV cell-mediated immunity assays have great potential as adjunctive tests for developing individualized management plans. These assays are useful for identifying patients who undergo immune reconstitution. It is crucial to conduct additional prospective interventional studies to gain a comprehensive understanding of their effectiveness [11].

In a study conducted by Heston et al., it was discovered that CMV viremia is prevalent among pediatric HCT recipients. The research revealed that certain factors, such as age, gender, race, type of transplant, and CMV seropositivity, can increase the risk of CMV viremia in children. Based on the study’s findings, it appears that children who develop CMV viremia within the first 100 days after HCT do not face an increased risk of all-cause mortality at the 2-year mark after HCT. This is despite our current institutional practices, such as preemptive therapy for CMV viremia [12]. Based on a concise and well-written small cohort study of pediatric HCT recipients, it was discovered that several factors, including older age, leukemia, allogeneic HCT, exposure to anti-thymocyte globulins (ATG), and acute GVHD, were linked to a higher incidence of CMV infections. However, upon further analysis, it was found that these factors did not have a direct correlation with CMV viremia [13].

### 1.3. Bidirectional Relationship between CMV and GVHD

Over the course of 60 years, extensive research has transformed allogeneic HCT from a therapy that was once considered obsolete in the 1960s to a now widely accepted treatment for life-threatening malignant and non-malignant blood diseases [14]. CMV infection and acute GVHD are significant complications that can greatly impact the prognosis following HCT. Early immune reconstitution (IR) is linked to better survival rates following HCT. Further research is needed to determine the effects of CMV infection, aGVHD, and IR when they occur together during HCT [15].

There is a clear association between GVHD and CMV replication. Studies have shown that patients undergoing GVHD treatment are more susceptible to CMV replication [11,16]. The role of CMV replication as a cause of GVHD is a topic of debate. Many review articles on the topic suggest a potential link between CMV replication and GVHD [17]. Additionally, CMV infection may heighten the likelihood of GVHD. According to reports, cells infected with CMV stimulate the production of IL-6, which can contribute to the development of GVHD [18]. The activation and proliferation of T cells, which play a crucial role in driving the immune response against the host tissues in GVHD, are promoted by IL-6. Studies have shown a clear connection between CMV-positive serology and the development of GVHD, resulting in increased mortality rates and decreased overall survival [11,15,19,20,21,22].

## 2. Cytomegalovirus Diagnosis

### 2.1. CMV Quantitative Nucleic Acid Test

CMV viral load is frequently measured using QNAT (quantitative nucleic acid test) to evaluate for CMV viremia and disease. The CMV viral load and its kinetics are strong indicators of disease progression and closely correlate with symptom resolution and treatment outcomes [23]. In individuals with weakened immune systems, the way the symptoms appear may be affected by various factors related to the person’s health and the virus itself. Several factors are important to consider: the type of infection (primary, reactivation, or superinfection), the specific transplant setting (solid-organ transplant or HCT), and the level of immunosuppression [24,25,26]. These factors have a significant impact.

Various preventive strategies have led to a decrease in the incidence of CMV disease and the associated short-term attributable mortality [27]. Transplant recipients are greatly affected by CMV, which has a notable negative impact. This is due to both direct high-grade viral replication, which leads to CMV disease and tissue injury, and indirect effects mediated by CMV that negatively affect transplant outcomes.

It is still difficult to establish the right thresholds to distinguish between CMV pneumonia and pulmonary shedding in HCT [28,29,30]. Gastrointestinal (GI) disease may not result in significant viremia in plasma or whole blood. Histopathology on GI biopsy samples is the gold standard for diagnosing CMV GI disease. A GI biopsy under endoscopy is the gold standard for diagnosing CMV gastroenteritis and aGVHD. However, it is an invasive procedure that carries the risk of severe side effects [31]. A recent retrospective analysis revealed promising findings regarding the effectiveness of quantitative CMV polymerase chain reaction (PCR) from GI tissue when compared to immunohistochemistry. However, further research is necessary to determine the appropriate thresholds and validate these findings on a larger scale [32].

### 2.2. Antigen for Cytomegalovirus

The presence of the CMV phosphoprotein 65 (pp65) antigen can be identified in peripheral blood leukocytes when someone has an active CMV infection [33]. The CMV antigen assay utilizes a monoclonal antibody for the detection of the pp65 antigen. Nevertheless, this test can be quite time-consuming, needs to be processed promptly, and lacks uniformity. Due to its inability to detect the viral protein in leukocytes, it does not provide any benefit for neutropenic patients [34]. For leukopenic patients, it is generally recommended to use CMV QNAT instead of the antigen test.

### 2.3. Culture and Histopathology

There are different methods available for viral culture, such as conventional and shell vial assays [35,36]. The effects on human fibroblasts are evaluated using traditional tests. Shell vial assays are capable of detecting antibodies to viral antigens. Both tests have reduced sensitivity and require longer processing times. Histopathology and immunohistochemistry are commonly used to diagnose invasive CMV disease. They provide highly specific results and are considered to be the gold standard.

## 3. Cytomegalovirus Prevention

As previously stated, CMV serostatus plays a crucial role in predicting the risk of post-transplant CMV reactivation and transplant-related morbidity and mortality because of its impact on the immune system. CMV-specific IgM and IgG antibodies are utilized to determine serostatus. There are two main approaches to preventing CMV infection: antiviral prophylaxis and preemptive therapy. In the HCT population, the preference has typically been for a preemptive treatment approach rather than antiviral prophylaxis. This is to avoid the potential for drug-induced toxicity, including the risk of bone marrow suppression from antivirals.

An effective treatment strategy involves regular screening for CMV viral load using PCR and initiating antiviral medication upon detection of viremia surpassing a specific threshold. This helps to prevent the development of end-organ disease. The threshold may vary between centers based on the specific CMV assay and the patient’s risk factors. Preemptive therapy has several benefits, including minimizing drug toxicity and costs, as well as potentially supporting early immune reconstitution by controlling CMV replication. Regular CMV PCR monitoring and patient adherence to laboratory visits are necessary. Various trials have compared different antiviral prophylaxis options, including acyclovir, valacyclovir, valganciclovir (VGCV), brincidofovir (BCV), and maribavir (MBV). These trials have shown a significant decrease in CMV disease but no significant differences in mortality rates [37,38]. During high-dose acyclovir prophylaxis, subclinical reactivations may occur, potentially leading to enough antigen stimulation. This stimulation is linked to the earlier reconstitution of specific immunity [4,39]. Antiviral prophylaxis, including valganciclovir and acyclovir, may lead to side effects such as cytopenias, which can heighten the risk of bacterial and fungal infections [40]. Therefore, many centers implemented preemptive therapy as a standard practice until the approval of letermovir (LET) in 2017.

### Letermovir

LET is currently the only drug recommended for CMV prophylaxis. Blocking the CMV viral terminase complex effectively stops the replication of the virus. A randomized controlled trial compared LET to standard-of-care preemptive therapy for a duration of 14 weeks. The study findings demonstrated a notable decrease in CMV infections after 24 weeks in the group that received prophylaxis (37.5% vs. 60%, *p* < 0.001), which also included the high-risk subgroup [41]. After 48 weeks, there were no significant changes in side effects. At 24 weeks, the LET group showed a lower all-cause mortality rate of 10.2% compared to 15.9% in the control group (*p* = 0.03). Similarly, at 48 weeks, the LET group had a lower all-cause mortality rate of 20.9% compared to 25.5% in the control group (*p* = 0.12). Patients who had clinically significant CMV events had a higher incidence of all-cause mortality compared to those without such events. The difference in mortality rates was statistically significant (31% vs. 18%; *p* = 0.02). There was no significant difference in all-cause mortality between patients with or without clinically significant CMV events in the LET group [42].

After the trial’s results, many centers have now adopted LET prophylaxis to prevent CMV disease after transplantation. The trial results have been successfully replicated in real-world settings. A study was conducted on a group of 53 individuals who were given LET prophylaxis for a duration of 14 weeks following an allogeneic transplant [11]. This study concluded that LET successfully prevented CMV infection. A considerable number of individuals in the study population were at a heightened risk of CMV reactivation due to either receiving a T cell-depleted graft or a haploidentical donor. Reactivation of CMV infection occurred in 5% of patients who were CMV R+. Out of the 29 patients who received extended prophylaxis beyond 14 weeks, only 3.4% experienced reactivation. Another study examined 29 patients who underwent CMV R+ Allo-HCT and compared those who received prophylaxis with those who did not. At 100 days, the LET group had a much lower incidence of clinically significant CMV infection compared to the control group. Only 4% of the LET group experienced the infection, while the control group had a much higher rate of 59% [43]. Many centers have adopted LET prophylaxis for high-risk patients. Nevertheless, the use of this treatment for CMV infection has not received approval and is currently not recommended due to the risk of failure resulting from its low resistance barrier.

Additionally, LET can be utilized as a secondary prophylaxis to prevent delayed CMV reactivation, specifically in individuals at high risk. Lin et al.’s study found that CMV reactivation was not observed in the 14 patients who received secondary prophylaxis with LET [44]. LET was administered as secondary prophylaxis in 80 Allo-HCT CMV R+ patients with a high risk of CMV reactivation in a compassionate study program in France. Factors that increase the risk include having a donor who is unrelated or haploidentical, using T cell-depleting agents, experiencing acute or chronic GVHD, and undergoing a cord blood transplant. Out of the patients included in this study, a majority (60%) utilized T cell-depleting agents such as alemtuzumab or ATG. Additionally, a significant proportion (67%) experienced GVHD in various stages. Grade 3–4 GVHD was observed in twenty-two percent of the patients. Out of the total of 80 cases, a small percentage of four individuals (5.5%) experienced breakthrough CMV infections. Additionally, one person developed resistance to LET, and unfortunately, there were six reported deaths [45]. Additional future studies are required. A phase 3 randomized clinical trial is currently underway to evaluate the extension of LET prophylaxis beyond day 100 in Allo-HCT. The trial is registered under the ClinicalTrials.gov Identifier: NCT03930615.

## 4. Immune Reconstitution after Allogeneic Hematopoietic Cell Transplantation

After Allo-HCT, neutrophils typically reconstitute within 2–3 weeks, with natural killer cells and T cells followed by day 100. Cellular immunity deficiency heightens the risk of viral infection reactivation. It may take 1–2 years for humoral immunity to fully recover.

CMV reactivation is controlled by CD4+ and CD8+ T helper cells, with a specific focus on their role. At different stages of maturation, they produce a range of cytokines, including interferon γ, IL-2, and TNF α. These cytokines likely play a role in controlling the CMV infection [3,46,47]. Having a CMV-specific T-cell response is crucial in preventing late CMV disease and mortality [48]. Zhang et al. discovered that CMV-specific CD8+ T cells are highly effective in diagnosing CMV reactivation in patients with HCT. An established threshold of 925 CMV-specific CD8+ T-cell counts per 106 PBMCs serves as an indicator of CMV reactivation after HCT. This threshold provides accurate guidance for prompt medication and improved management of CMV infection following HCT [49]. Extended CMV and CMV-targeted treatment can potentially hinder the restoration of virus-specific immune responses, increasing the risk of disease occurring later on [50]. Several factors can contribute to delays in immune reconstitution (IR). These include the specific conditioning regimen used, the use of steroids, the occurrence of GVHD, transplants from HLA-mismatched or unrelated donors, the use of bone marrow as the source of stem cells, and the administration of ganciclovir (GCV) prophylaxis [51,52,53]. New treatments for GVHD, such as ruxolitinib, have the potential to impact T cell function and potentially prolong the process of IR [54].

### Cytomegalovirus Cell-Mediated Immunity Assays

CMV significantly contributes to morbidity and mortality in pediatric transplantation. Existing prevention strategies have their limitations, prompting researchers to explore new strategies. Cell-mediated immunity (CMI) plays a vital role in managing CMV infection, and CMV-specific CMI assays hold potential for both prevention and treatment [55].

CMV CMI assays detect the expression of IFN-γ by CD4+ and/or CD8+ cells following stimulation with CMV antigens. Assays commonly utilize peptide pools from immediate early protein-1 and pp65 that overlap [56,57]. Additionally, it is worth noting that the human immune response to CMV extends beyond these proteins, and CMI assays may not accurately identify all patients with measurable CMV-specific immunity [58]. No commercial CMV CMI assay has received approval from the Food and Drug Administration in the United States. However, several assays are licensed for use in Europe [59]. An ELISA-based method is used to detect the production and secretion of IFN-γ by T cells in whole blood after being stimulated with CMV peptides [60]. The ELISpot assays can detect the presence of IFN-γ near activated CD8+ and CD4+ T cells after T cell stimulation in microtiter plates [61]. Flow cytometry-based assays can measure IFN-γ expression by utilizing intracellular cytokine staining [62].

The timing of T cell reconstitution after transplantation can impact CMV CMI. The CMV-specific T-cell response in patients with solid organ transplantation (SOT) decreases over time. The CD8 + T cell response declines early but returns to pretransplant levels within 2 months. The CD4 + T cell response is at its lowest point around 2 months after transplant and takes about 12 months to return to pretransplant levels. The induction immunosuppression regimen chosen greatly affects T cell reconstitution in SOT. Additionally, valganciclovir’s strong suppression of viral replication may hinder CMV CMI in seronegative recipients [63].

## 5. Cytomegalovirus Treatment

Although transplantation has seen rapid progress, the development of CMV therapeutics has not kept pace. Only six CMV therapeutics have been approved by the US Food and Drug Administration (FDA). Foscarnet (FOS) was approved in 1991 [64], followed by ganciclovir (GCV) [65] and its prodrug valganciclovir (VGCV) [66] in 1994 and 2001, respectively. Cidofovir (CDV) was approved in 1996 [67], LTV in 2017 [68], and MBV in 2021 [69]. Each of these medications has antiviral properties for CMV, although their efficiency against other viruses, such as herpes simplex and varicella zoster, differs. Table 1 delineated the treatment and prevention of various antiviral agents for cytomegalovirus.

### 5.1. Ganclovir and Valganciclovir

GCV is converted into its active triphosphate form inside the body, which inhibits the replication of CMV DNA [70]. Both intravenous and oral forms of the drug are available, but the oral form is not very effective because it has low bioavailability. VGCV is a form of GCV that can be taken orally. A trial comparing VGCV with intravenous GCV showed similar effectiveness and safety profiles [71]. Another study found that the rates of viremia clearance are comparable to VGV and GCV [72]. VGCV demonstrated comparable effectiveness to GCV in decreasing CMV viral load in individuals with T cell-depleted allografts [73].

### 5.2. Foscarnet

FOS selectively inhibits viral polymerase and is only available in an intravenous formulation. It is a pyrophosphate analog. A study conducted at multiple centers compared the effectiveness and survival rates of GCV and FOS. The results showed that both treatments had similar efficacy and survival rates at 180 days. Additionally, the FOS treatment had the advantage of causing less hematotoxicity [74]. This suggests that FOS may be a preferable option in terms of safety. It may be the preferred option for preemptive therapy in cases where marrow toxicity is a concern, particularly during the pre-engraftment phase and in instances of GCV resistance. FOS has the potential to cause nephrotoxicity by directly damaging the renal tubular cells. It can also lead to electrolyte imbalances, particularly potassium and magnesium, which necessitate careful monitoring. Avoid in patients with or at risk of renal disease.

### 5.3. Marbavir

MBV is an antiviral drug that works by blocking the activity of a viral protein kinase called unique long (UL) 97, which ultimately stops the virus from replicating. Response rates at 6 months were similar (79% vs. 67%) in a phase 2 trial comparing different doses of MBV with VGCV in Allo-HCT and solid organ transplant patients [75]. In the MBV group, a significant number of participants experienced gastrointestinal side effects, particularly dysgeusia, which resulted in the discontinuation of the drug for 23% of them. In comparison to VGCV, the occurrence of neutropenia was significantly lower (6% vs. 22%). During a phase 2 randomized controlled trial, MBV was tested at various doses for refractory and resistant CMV infections [76]. The study found that a significant number of patients, ranging from 63% to 70%, achieved an undetectable CMV viral load within 6 weeks. However, 25 patients (20%) experienced recurrent infections, with 13 of them developing resistance mutations to MBV. No significant marrow or renal toxicity was reported in any of these studies involving MBV. Studies have indicated that MBV is a promising oral medication for preemptive therapy and the treatment of viral infections that are resistant to other drugs. It has been found to have minimal toxic effects on the bone marrow. However, the long-term use of MBV may be restricted due to the possibility of developing drug resistance.

### 5.4. Cidofovir

CDV is a nucleotide analog that gets converted into its diphosphate form and blocks viral DNA polymerase. It can be found in IV form and is usually given once a week. Its usage is restricted because it poses a significant risk of nephrotoxicity, which can cause damage to the tubules. Administering it with saline and probenecid can decrease its renal excretion, reducing the dosage required. Additionally, it may lead to a decrease in bone marrow function and potential side effects on the eyes. Some studies have found that patients who were treated with CDV for CMV retinitis experienced ocular side effects, including uveitis, iritis, and hypotonia [77,78]. Both intravitreal and intravenous formulations were observed. Consider its use in managing resistant CMV infections when no other drug options are available.

### 5.5. Brincidofovir

BCV is an orally administered prodrug of CDV that is conjugated with lipids. Minimizing the harmful side effects of CDV on the kidneys and bone marrow is beneficial. The results of a phase 3 randomized controlled trial comparing BCV to placebo for CMV prophylaxis in Allo-HCT patients showed no significant reduction in CMV infections at 24 weeks. The drug was not approved by the FDA for this indication due to the higher rates of diarrhea and GVHD [79].

VGCV and GCV are the preferred and first-line agents for treating CMV viremia and infection. Unfortunately, these drugs have a significant drawback: they can cause bone marrow suppression. We opt for FOS as the preferred treatment for early CMV reactivation during the pre-engraftment phase. This helps avoid any potential marrow toxicity that may arise from valganciclovir and ganciclovir. We typically administer treatment for a minimum of two weeks or until the CMV viremia is no longer present, whichever duration is longer [80].

### 5.6. Cytomegalovirus Immune Globulin

CMV immune globulin (CMV Ig) is a commercially available product purified from pooled adult human plasma selected for high antibody titers against CMV. Its application could provide some degree of immunity against CMV reactivation or disease. Numerous studies have advocated for the use of CMV Ig after HCT, especially in high-risk cadaveric allograft recipients. Recently, the FDA expanded the approved indications of CMV Ig to include prophylaxis against CMV disease associated with lung, liver, pancreas, and heart transplantation. However, prophylactic CMV Ig should be combined with GCV. Despite the advantage of infrequent dosing, CMV Ig may not be cost-effective. Therefore, in most transplant centers, CMV Ig in combination with IV GCV is reserved for treating invasive CMV disease in high-risk patients.

## 6. Cytomegalovirus Antiviral Drug Resistance

Refractory CMV infection is characterized by an increase in CMV viral load (>1 log) despite receiving the correct dosage of treatment for 2 weeks. On the other hand, resistant CMV infection is identified by persistent viremia, along with the presence of genotypic drug resistance mutations [80]. CMV drug resistance mutations have been reported in Allo-HCT patients, with prevalence ranging from 0 to 8% [81,82]. A study found that 14% of haploidentical HCT patients who had been on prolonged antiviral treatment (median = 70 days) developed resistance [83]. Immunologic failure is typically the primary cause of CMV control issues in the HCT population, rather than drug resistance. Resistance to ganciclovir, valganciclovir, and MBV is caused by mutations in the UL97 kinase gene [84]. Patients with UL97 kinase mutations and less than 5-fold GCV resistance, without end-organ disease, may benefit from an increased dose of 7.5–10 mg/kg of GCV. Monotherapy with FOS is recommended for mutations that confer resistance greater than 5-fold. The UL54 gene mutations affect the viral DNA polymerase enzyme, leading to resistance against FOS, cidofovir, ganciclovir, valganciclovir, and MBV. The resistance pattern would determine the most suitable treatment for these cases [85]. Figure 1 depicts human CMV-infected cells and the mechanisms of action of antiviral therapies. HCT has a higher prevalence of UL97 gene mutations compared to UL54 gene mutations.

## 7. Use of Adoptive T-Cell Therapy as Prophylaxis

Given the importance of cell-mediated immunity in managing stubborn viral infections, the utilization of T cells as a treatment option is becoming increasingly appealing. Controlling resistant and refractory CMV infections can be quite useful. Generating virus-specific T cells can be achieved by stimulating virus-specific cells with a viral protein. These cells have the potential for in vivo use to expand further or for direct infusion into the recipient [86]. Various methods can be used to isolate T cells from a CMV seropositive matched donor, including HLA class multimers and interferon γ capture. Acquiring these cells from donors can take 4–6 weeks, which makes this procedure unsuitable for quickly treating severe CMV disease. Now, banks are being created using third-party donors to offer ready-to-use products [87]. GVHD caused by HLA mismatches from third-party products is a potential concern. Nevertheless, the frequency of occurrence has been relatively low in published studies [88,89,90]. Previous studies on adoptive T-cell therapy for refractory CMV infections had small sample sizes and lacked randomized controlled trials with comparison groups [18,91,92,93]. Further research is necessary to assess its effectiveness and potential side effects.

Research has indicated that adoptive T-cell therapy may have the potential to prevent CMV infections. In a study involving 50 patients, it was found that a CMV T cell infusion given 28 days after transplant resulted in CMV reactivation in 26 individuals. Notably, five cases of reactivation occurred specifically after the infusion. Nine underwent therapy using antivirals. In the treated group, a lower percentage of patients (17% vs. 36%, *p* = 0.01) required antiviral therapy. GVHD did not increase, and the OS rates were comparable in both groups [94].

## 8. Cytomegalovirus Vaccines

There are currently multiple ongoing vaccine trials aimed at preventing CMV infection. Transvax, developed by Vical, is a vaccine that includes plasmids encoding pp-65 and glycoprotein B (gB). It enhances the body’s immune response to both proteins. The development of a safe and effective CMV vaccine is a major medical priority. Several studies are ongoing, focusing on the gB of CMV. ASP0113, a bivalent DNA vaccine, has shown significant reductions in CMV viremia occurrence and recurrence. A different vaccine, made from soluble recombinant gB with adjuvant MF59, effectively decreased the duration of viremia and the number of days of GCV treatment required. CSJ148, a trial for CMV monoclonal antibodies, is currently under review [95].

Vaccination against CMV infection is crucial, and gB has shown partial protection in clinical trials. However, poor neutralizing responses against the antigenic domains of gB were found. A new antibody, AD-6, is detected in 70% of vaccine recipients but not in 5% of naturally infected people, potentially explaining some of the protection mediated by gB vaccines [96]. PepVax, created by Helocyte, is a peptide vaccine containing an HLA-restricted CD8 T cell epitope from the pp65 protein and a Toll-like receptor 9 agonist as an adjuvant. It boosts the body’s cellular immune response. A phase 1 trial showed a decrease in CMV reactivation and the requirement for antivirals, as well as a doubling of CD 8+ T cell immunity [97]. The initial trial showed promising results, and a follow-up trial is currently underway. Triplex is a viral vector-based (Ankara) vaccine that has shown safety and tolerability in a phase 1 trial [98]. The development of the CMV vaccine shows promise, but it seems that it will take several years to transition to a clinical setting.

A study involving 402 CMV seronegative girls aged 12–17 years showed that the CMV gB vaccine was generally well-tolerated, with local and systemic adverse events more common in the vaccine group [99]. The vaccine induced gB antibodies in all recipients, and 48 CMV infections were detected. The vaccine was safe and immunogenic, consistent with a previous study in adult women using the same formulation [100]. Another study comparing the effectiveness of a messenger RNA-based vaccine candidate, mRNA-1647, against human CMV found that mRNA-1647 induced polyfunctional and durable CMV-specific antibody responses [101]. The phase 1 trial of mRNA-1647, an mRNA-based CMV vaccine, evaluated its safety, reactogenicity, and immunogenicity in adults. The trial involved 154 participants, with 118 receiving mRNA-1647 and 36 receiving placebo. Results showed no deaths, serious adverse events, or special interest events. Most adverse reactions were of grade ≤ 2 severity. The trial concluded that mRNA-1647 has an acceptable safety profile in adults. Despite lower gB-specific IgG responses, mRNA-1647 elicited higher neutralization and antibody-dependent cellular cytotoxicity responses compared to the gB/MF59 vaccine. This suggests that improving the design of the vaccine could lead to a suitable vaccine for licensure [102].

This review highlights current knowledge, identifies gaps, and suggests future research directions. It aids researchers in understanding existing literature and potential areas for further study. The authors also analyze self-reported limitations, guiding future investigations.

## 9. Conclusions

Improved strategies and drugs are urgently needed to prevent and treat CMV infections, as they significantly increase the risk of mortality, morbidity, and GVHD in Allo-HCT patients. We need more studies to evaluate how useful CMV-CMI assays are. A prospective trial is currently investigating the potential benefits of extended prophylaxis with LET beyond day 100 [103]. LET acts by targeting the CMV terminase complex, which inhibits the cleavage and packaging of viral DNA into capsids. MBV disrupts the process of capsid assembly and the release of virions. The drugs are effective against most CMV strains that are resistant to standard drugs, although there is a possibility of high-level resistance mutations occurring more quickly. Adoptive T-cell therapy is a promising approach for addressing stubborn CMV infections. However, more extensive randomized trials are needed to further assess its effectiveness.

## Figures and Tables

**Figure 1 viruses-16-01268-f001:**
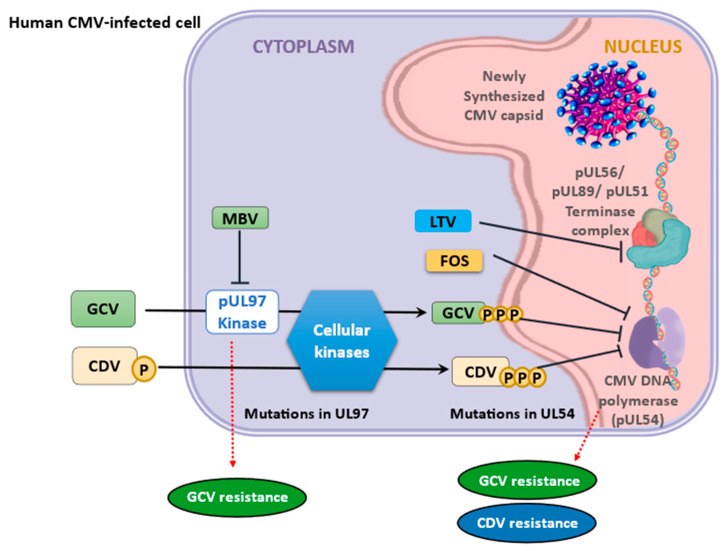
Human CMV-infected cells and mechanisms of action of antiviral therapies. Abbreviations: CDV, cidofovir; CMV, cytomegalovirus; FOS, foscarnet; GCV, ganciclovir; LTV, letermovir; MBV, maribavir. The figure is primarily derived from studies by Saullo et al. (2023) [70] and Panda et al. (2023) [85].

**Table 1 viruses-16-01268-t001:** Antiviral medications used to prevent and treat cytomegalovirus in children undergoing hematopoietic stem cell transplantation.

Agent	Indications in Transplant Recipients	Dosing for CMV Prevention in Pediatric Transplant	Dosing for CMV Treatment in Pediatric Transplant	Adverse Effects	Major Resistance Mutations
Valganciclovir	First line for treatment and prevention of CMV DNAemia and disease	7 × BSA × CrCLS with an upper limit of creatinine clearance of 150 mL/min PO daily (max dose 900 mg PO daily) or 16 mg/kg PO daily (max dose 900 mg PO daily)	7 × BSA × CrCLS with an upper limit of creatinine clearance of 150 mL/min PO BID (max dose 900 mg PO BID) or 16 mg/kg PO BID (max dose 900 mg PO BID)	Hematologic toxicity, infertility, fetal toxicity, nephrotoxicity, diarrhea	UL97 > UL54
Ganciclovir	First line for treatment and prevention of CMV DNAemia and disease	5 mg/kg IV once daily	5 mg/kg IV every 12 h	Hematological toxicity, infertility, fetal toxicity, nephrotoxicity, diarrhea, headache	UL97 > UL54
Foscarnet	Second-line agent for therapy in SOT recipients First-line agent for HCT recipients who cannot tolerate marrow suppression from Ganciclovir/Valganciclovir	Currently not indicated for prevention in SOT recipients 90 mg/kg IV every 24 h for prevention in HCT	Induction: 60 mg/kg/dose q8h or 90 mg/kg q12 h for 7–14 days. Maintenance: 90 mg/kg/dose once daily until the indicator test is negative. Renal dose adjustment is required; no hepatic adjustment is needed	Seizures, nephrotoxicity, headache, hypokalemia, hypocalcemia, hypomagnesemia, hypophosphatemia, nausea, vomiting, diarrhea, anemia, granulocytopenia	UL54
Cidofovir	A third-line drug used to treat CMV DNAemia and disease in SOT patients second-line drug for treating and preventing CMV DNAemia and disease in people who have had a heart transplant Second-line treatment for CMV DNAemia or illness that is resistant to ganciclovir UL97 mutation	Not recommended for prevention in SOT users at this time 5 mg/kg/dose IV once a week for two weeks in a row, then 5 mg/kg/dose IV once every two weeks, along with probenecid in HCT	5 mg/kg/dose IV once a week for two weeks in a row, then 5 mg/kg/dose IV once every two weeks, along with probenecid	Nephrotoxicity, neutropenia, carcinogenic and teratogenic, infusion reactions, headache, nausea	UL54
Maribavir	Refractory CMV DNAemia or disease (with or without resistance to traditional agents) age 12 and older	Currently not indicated for CMV prevention	400 mg PO twice daily for patients ≧12 years of age and ≧35 kg	Changes in taste, diarrhea, loss of taste, nausea, unusual tiredness or weakness, and vomiting. Recommend reviewing contaminant medications as it has numerous other drug-drug interactions Appropriate studies have not been performed on the relationship of age to the effects of MBV in children younger than 12 years of age.	UL97
Letermovir	Prophylaxis of CMV infection and disease in adult CMV-seropositive recipients [R+] of an allogeneic hematopoietic stem cell transplant	480 mg PO or IV administered once daily (over 1 h) through 100 days posttransplant; dose should be reduced to 240 mg if coadministered with cyclosporine	Currently not indicated for treatment	Headache, nausea, peripheral edema, diarrhea When coadministered with CYP3A4 inhibitors, including cyclosporine, tacrolimus, and statins, dose modifications and constant monitoring are needed.	UL56 > UL51 and UL89

Abbreviations: BID, twice daily; BSA, body surface area; CrCLS, serum creatinine clearance; CMV, cytomegalovirus; HCT, hemopoietic cell transplantation; IV, intravenous; kg, kilograms; mg, milligrams; mL, milliliter; min, minute; PO, orally; SOT, solid organ transplantation; UL, unique long.

## Data Availability

Not applicable.

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
