# Peer review of "Antiviral Agents for Preventing Cytomegalovirus Disease in Recipients of Hematopoietic Cell Transplantation"

_viruses, 2024, doi:10.3390/v16081268_

Round 1

Reviewer 1 Report

Comments and Suggestions for Authors

I would like to thank the author for his submission and allowing me to review his work.

This is an interesting study on an important topic. However, I would be grateful if he could add further explanations and changes on the following points:

1) ABSTRACT: Page 1, line 11

What type of review is it? (Narrative? Systematic? Other?). Please, clarify.

2) INTRODUCTION: Page 1, line 28

In the introduction section, I suggest including more information about CMV. Although it is a common infection, it generally presents as asymptomatic or paucisymptomatic. However, in immunocompromised individuals or children with congenital infection, it can lead to several complications, such as sensorineural hearing loss (Reference: Congenital Cytomegalovirus and Hearing Loss: The State of the Art. J Clin Med. 2023;12(13):4465. doi:10.3390/jcm12134465)

3) INTRODUCTION: Page 1, line 41

I suggest specifying the type of study, stating the aim of the study, and describing how it was conducted (for example, by screening the abstracts of relevant literature).

4) INTRODUCTION: Page 2, line 91

Please provide references of the “numerous studies” mentioned in the text.

5) INTRODUCTION: Page 2, line 95

The role of IL-6 in GVHD pathogenesis is mentioned but could be expanded.

6) CYTOMEGALOVIRUS PREVENTION: Page 5, line 206

Acronyms need to be explained only the first time they appear in the text.

7) CYTOMEGALOVIRUS VACCINES: Page 10, line 395

It would be beneficial to include more information about the specific results of the trials, such as the number of participants, the duration of the trials, and any side effects observed.

8) Page 10, line 395

What are the limitations and future prospects of this article?

Comments on the Quality of English Language

Moderate editing is required

Author Response

Manuscript: viruses-3133934

The manuscript has been amended based on the reviewer's comments. To avoid plagiarism, we rephrased the sentences in this updated paper while citing the source. The red highlights identify all of the modifications, and we respectfully urge that the reviewer reconsider the updated paper.

Reviewer 1: I would like to thank the author for his submission and for allowing me to review his work.

This is an interesting study on an important topic. However, I would be grateful if he could add further explanations and changes on the following points:

1) ABSTRACT: Page 1, line 11

What type of review is it? (Narrative? Systematic? Other?). Please, clarify.

Response: The sentence in the introduction section has been revised: “This systematic review discusses the use of prophylaxis to prevent cytomegalovirus (CMV) infection in recipients who have undergone hematopoietic cell transplantation.”

2) INTRODUCTION: Page 1, line 28

In the introduction section, I suggest including more information about CMV. Although it is a common infection, it generally presents as asymptomatic or paucisymptomatic. However, in immunocompromised individuals or children with congenital infection, it can lead to several complications, such as sensorineural hearing loss (Reference: Congenital Cytomegalovirus and Hearing Loss: The State of the Art. J Clin Med. 2023;12(13):4465. doi:10.3390/jcm12134465)

Response: Additional information about CMV has been included for further reading: “Despite its prevalence, this infection typically manifests as asymptomatic or pauci-symptomatic. Nevertheless, it can result in a variety of complications, including sensorineural hearing loss, in immunocompromised individuals or infants with con-genital infections [1].” We have included the appropriate reference in reference 1, as you suggested.

3) INTRODUCTION: Page 1, line 41

I suggest specifying the type of study, stating the aim of the study, and describing how it was conducted (for example, by screening the abstracts of relevant literature).

Response: Based on your suggestion, we have made revisions to the introduction and it now reads: “The objective of this study was to conduct a comprehensive review of the latest antiviral agents by analyzing current literature.” 

4) INTRODUCTION: Page 2, line 91

Please provide references to the “numerous studies” mentioned in the text.

Response: The pertinent citations were referenced as 14 and 15, as demonstrated below:

  1. Cantoni, N.; Hirsch, H.H.; Khanna, N.; Gerull, S.; Buser, A.; Bucher, C.; Halter, J.; Heim, D.; Tichelli, A.; Gratwohl, A.; et al. Evidence for a bidirectional relationship between cytomegalovirus replication and acute graft-versus-host disease. Biol. Blood. Marrow. Transplant. 2010; 16:1309-1314.
  2. Jakharia, N.; Howard, D.; Riedel, D.J. CMV Infection in Hematopoietic Stem Cell Transplantation: Prevention and Treatment Strategies. Curr. Treat. Options. Infect. Dis. 2021; 13:123-140.

5) INTRODUCTION: Page 2, line 95

The role of IL-6 in GVHD pathogenesis is mentioned but could be expanded.

Response: Thank you sincerely for your perceptive remarks and valuable recommendations. The paragraph has been edited and now appears in its amended form: Additionally, CMV infection may heighten the likelihood of GVHD. According to reports, cells infected with CMV stimulate the production of IL-6, which can contribute to the development of GVHD [17]. The activation and proliferation of T cells, which play a crucial role in driving the immune response against the host tissues in GVHD, are promoted by IL-6.

6) CYTOMEGALOVIRUS PREVENTION: Page 5, line 206

Acronyms need to be explained only the first time they appear in the text.

Response: We made the suggested revisions to the acronyms.

7) CYTOMEGALOVIRUS VACCINES: Page 10, line 395

It would be beneficial to include more information about the specific results of the trials, such as the number of participants, the duration of the trials, and any side effects observed.

Response: Comprehensive details regarding the individual outcomes of the experiments are presented in lines 419-434, which now state: “A study involving 402 CMV seronegative girls aged 12-17 years showed that the CMV gB vaccine was generally well-tolerated, with local and systemic adverse events more common in the vaccine group [92]. The vaccine induced gB antibody in all recipients and 48 CMV infections were detected. The vaccine was safe and immunogenic, consistent with a previous study in adult women using the same formulation [93]. Another study comparing the effectiveness of a messenger RNA-based vaccine candidate, mRNA-1647, against human CMV, found that mRNA-1647 induced polyfunctional and durable CMV-specific antibody responses [94]. The phase 1 trial of mRNA-1647, an mRNA-based CMV vaccine, evaluated its safety, reactogenicity, and immunogenicity in adults. The trial involved 154 participants, with 118 receiving mRNA-1647 and 36 receiving placebo. Results showed no deaths, serious adverse events, or special interest events. Most adverse reactions were grade ≤2 severity. The trial concluded that mRNA-1647 has an acceptable safety profile in adults. Despite lower gB-specific IgG responses, mRNA-1647 elicited higher neutralization and anti-body-dependent cellular cytotoxicity responses compared to the gB/MF59 vaccine. This suggests that improving the design of the vaccine could lead to a suitable vaccine for licensure [95].”

8) Page 10, line 395

What are the limitations and future prospects of this article?

Response: We added a sentence that is relevant to the topic: This review highlights current knowledge, identifies gaps, and suggests future research directions. It aids researchers in understanding existing literature and potential areas for further study. The authors also analyze self-reported limitations, guiding future investigations.”

incorporated the various information categories into the revised manuscript, as per your advice. We kindly request your understanding and acceptance of the required modifications to move forward with the publication of the manuscript. Additionally, an English-native speaker reviewed and made revisions to this manuscript.

Reviewer 2 Report

Comments and Suggestions for Authors

This is a wellcome review that well covers antiviral agents used in hematopoietic cell transplantation for treatment and prevention of CMV infection..

However,some editing is needed to improve the paper.

Regarding the references,these should be given after the first sentence  when referring to a study and not later in the chapter.This is a problem throughout the whole review.

On line 45 is stated that CMV seronegative donors/seropositive recipients have the highest rate of survival.This is wrong and should be corrected.The best survival is seen in CMV sero neg donor/recipient pairs.This should  be mentioned with appropriate references.In CMV seropositive patients there are studies suggesting that patienets with a seropositive donors have a better outcome  vs those who have a CMV seronegative donor.This should also be mentioned.The statement on line 59 is therfore debateable.

I miss some early references when Gancvyclivir and Foscarnet were introduced as therapy for transplant recipients with CMV disease.

Abbreviations used should be defined at the first time they appear.This is not the case in many places and somtimes abbreviations are explained several times for GVHD L38,L207 etc

Regarding HCT L 33 and L 35 etc!

On L 233-235 regarding Ruxolitinib ,please give a reference.

Repetitions should be avoided an example L237.

L263-269 regarding approvals of the drugs please give references for insatance of early studies the approvals were based on or references to approvals.

To conclude.This is an important review regarding therapeutics for CMV in HCT patients.Some editing is needed before publication to improve the paper.

Comments on the Quality of English Language

The article should be scrutinized by a native english speaking editor or colleague.Alternatively AI can be used to check the language for perfect american english.

Author Response

Reviewer 2: This is a welcome review that well covers antiviral agents used in hematopoietic cell transplantation for the treatment and prevention of CMV infection. However, some editing is needed to improve the paper.

  1. Regarding the references, these should be given after the first sentence when referring to a study and not later in the chapter. This is a problem throughout the whole review.

Response: Based on your suggestion, we have revised the reference numbers and placed them after the first sentence when mentioning the study.

  1. On line 45 is stated that CMV seronegative donors/seropositive recipients have the highest rate of survival. This is wrong and should be corrected. The best survival is seen in CMV sero neg donor/recipient pairs. This should be mentioned with appropriate references. In CMV seropositive patients there are studies suggesting that patients with seropositive donors have a better outcome than those who have a CMV seronegative donor. This should also be mentioned. The statement on line 59 is therefore debatable.

Response: Thank you for your feedback. We have revised the sentences accordingly. The paragraph has been revised and now has a more polished and refined tone. “CMV seronegative donor/recipient pairs exhibit the highest survival rates [5]. There have been studies indicating that CMV seropositive patients may experience better outcomes when receiving a transplant from a seropositive donor, compared to those who receive a transplant from a CMV seronegative donor [6].”

  1. I miss some early references when Gancyclovir and Foscarnet were introduced as therapies for transplant recipients with CMV disease.

Response: There was an early reference in a publication by Moretti et al in 1998 regarding bone marrow transplant. A study was conducted to compare the effectiveness of foscarnet and ganciclovir as pre-emptive therapy for CMV infection in patients undergoing allogeneic hemopoietic stem cell transplant. A total of 29 patients were randomly assigned to receive either foscarnet or ganciclovir for a duration of 15 days. The primary endpoints of the study focused on the outcome of CMV antigenemia, progression to CMV disease, and any potential side effects of the treatment. According to the study, foscarnet was more effective in clearing CMV antigenemia, although there were some failures in both the foscarnet and ganciclovir groups. The 1-year TRM for actuarial purposes was 25% compared to 12%. Foscarnet and ganciclovir are both effective for pre-emptive therapy of CMV antigenemia.

  1. Abbreviations used should be defined the first time they appear. This is not the case in many places and sometimes abbreviations are explained several times for GVHD L38, L207, etc; Regarding HCT L 33 and L 35, etc!

Response: Thank you for bringing this to my attention. The abbreviations have been corrected.

  1. On L 233-235 regarding Ruxolitinib, please give a reference.

Response: Following your advice, a pertinent citation was provided as reference 57.

  1. Repetitions should be avoided an example L237.

Response: Thank you for your feedback. We will refrain from using the example.

  1. L263-269 regarding approvals of the drugs please give references for insatance of early studies the approvals were based on or references to approvals.

Response: Thank you for the excellent proposal. References for FDA approval were provided appropriately. The paragraph was rewritten in the following way: “Although transplantation has seen rapid progress, the development of CMV therapeutics has not kept pace. Only six CMV therapeutics have been approved by the US Food and Drug Administration (FDA). Foscarnet (FOS) was approved in 1991 [67], followed by ganciclovir (GCV) [68] and its prodrug valganciclovir (VGCV) [69] in 1994 and 2001 respectively. Cidofovir (CDV) was approved in 1996 [70], LTV in 2017 [71], and MBV in 2021 [72] (see Table 1 and Figure 1). Each of these medications has antiviral properties for CMV, although their efficiency against other viruses such as herpes simplex and varicella zoster differs.”

  1. In conclusion: This is an important review regarding therapeutics for CMV in HCT patients. Some editing is needed before publication to improve the paper.

Response: Thank you for your insightful recommendation. I am extremely grateful. We have incorporated the various information categories into the revised manuscript, as per your advice. We kindly request your understanding and acceptance of the required modifications to move forward with the publication of the manuscript. Additionally, an English-native speaker reviewed and made revisions to this manuscript.

Round 2

Reviewer 1 Report

Comments and Suggestions for Authors

The Authors have clarified all my doubts.

Comments on the Quality of English Language

Good